# Enhanced Recovery after Surgery (ERAS) Implementation and Barriers among Healthcare Providers in France: A Cross-Sectional Study

**DOI:** 10.3390/healthcare12040436

**Published:** 2024-02-08

**Authors:** Augustin Clet, Marin Guy, Jean-François Muir, Antoine Cuvelier, Francis-Edouard Gravier, Tristan Bonnevie

**Affiliations:** 1Université Rouen Normandie, Normandie Univ, GRHVN UR 3830, F-76000 Rouen, France; jf.muir@adir-hautenormandie.com (J.-F.M.); f.gravier@adir-hautenormandie.com (F.-E.G.); t.bonnevie@adir-hautenormandie.com (T.B.); 2ADIR Association, Rouen University Hospital, F-76000 Rouen, France; antoine.cuvelier@chu-rouen.fr; 3Centre Aquitain Du Dos, Hôpital Privé Saint-Martin, F-33600 Pessac, France; maringuy.kine@gmail.com; 4Pulmonary, Thoracic Oncology and Respiratory Intensive Care Department, Rouen University Hospital, F-76000 Rouen, France

**Keywords:** ERAS, healthcare providers, practice, barriers

## Abstract

The implementation of Enhanced Recovery After Surgery (ERAS) is a challenge for healthcare systems, especially in case of patients undergoing major surgery. Despite a proven significant reduction in postoperative complications and hospital lengths of stay, ERAS protocols are inconsistently used in real-world practice, and barriers have been poorly described in a cohort comprising medical and paramedical professionals. This study aims to assess the proportion of French healthcare providers who practiced ERAS and to identify barriers to its implementation amongst those surveyed. We conducted a prospective cross-sectional study to survey healthcare providers about their practice of ERAS using an online questionnaire. Healthcare providers were contacted through hospital requests, private hospital group requests, professional corporation requests, social networks, and personal contacts. The questionnaire was also designed to explore barriers to ERAS implementation. Identified barriers were allocated by two independent assessors to one of the fourteen domains of the Theoretical Domains Framework (TDF), which is an integrative framework based on behavior change theories that can be used to identify issues relating to evidence on the implementation of best practice in healthcare settings. One hundred and fifty-three French healthcare providers answered the online questionnaire (76% female, median age 35 years (IQR: 29 to 48)). Physiotherapists, nurses, and dieticians were the most represented professions (31.4%, 24.2%, and, 14.4%, respectively). Amongst those surveyed, thirty-one practiced ERAS (20.3%, 95%CI: 13.9 to 26.63). Major barriers to ERAS practice were related to the “Environmental context and resources” domain (57.6%, 95%CI: 49.5–65.4), e.g., lack of professionals, funding, and coordination, and the “Knowledge” domain (52.8%, 95%CI: 44.7–60.8), e.g., ERAS unawareness. ERAS in major surgery is seldom practiced in France due to the unfavorable environment (i.e., logistics issues, and lack of professionals and funding) and a low rate of procedure awareness. Future studies should focus on devising and assessing strategies (e.g., education and training, collaboration, institutional support, the development of healthcare networks, and leveraging telehealth and technology) to overcome these barriers, thereby promoting the wider implementation of ERAS.

## 1. Introduction

Over the past three decades, major surgery has undergone remarkable changes, such as minimally invasive surgery, robot-assisted surgery, the management of anesthesia, analgesia, and sedation, as well as the integration of non-invasive ventilation (NIV) strategies. The emergence of Enhanced Recovery After Surgery (ERAS) protocols [1] is based on these surgical advances. Simultaneously, the field of surgery has encountered new challenges [2], including the reduction of postoperative complications and the minimization of physical stress on patients to speed up their return to work. ERAS protocols have emerged as a response to these advancements [3], encompassing an integrative surgical approach aimed at optimizing the entire perioperative period. This approach involves a comprehensive preoperative preparation, the reduction of stress during surgery, and the optimization of the postoperative period with a multidisciplinary strategy and targeted interventions to speed up the healing process. ERAS protocols are grounded in patient education and information, minimally invasive surgical procedures, and the coordinated management of a diverse team of medical, paramedical, and administrative professionals [4].

For instance, during the preoperative phase, this involves providing psychological and social support, managing risk factors such as smoking cessation, improving nutritional status, and implementing rehabilitation programs to enhance the participants’ physical fitness [5,6] while also educating patients about postoperative care [5,7]. In the perioperative phase, efforts are made to preempt postoperative complications (POC), nausea, vomiting, and pain, optimize intraoperative fluid management, and utilize minimally invasive surgical techniques [8,9]. The postoperative stage entails activities such as early oral intake, ileus prevention, pain management, weaning off mechanical ventilation, urinary drainage, potential chest physiotherapy, and early mobilization [1,8,9].

ERAS protocols must strike a delicate balance between delivering optimal care quality and reducing healthcare costs [10]. Consequently, the primary objectives include diminishing POC [11,12,13,14], reducing lengths of stay (LOS) [11,12,13,14,15,16,17], alleviating postoperative pain, and facilitating the swift return to normal physiological function. This approach has demonstrated its efficacy in enhancing the patients’ quality of life [18] and satisfaction [19] during the perioperative period. Initially aimed at general surgery, the adoption of ERAS protocols has rapidly expanded to encompass major surgical procedures [1,4]. Surprisingly, despite substantial supportive evidence, the ERAS Society has only certified 107 health facilities worldwide [7], and the actual number of healthcare providers who practice ERAS remains inadequately explored. Nevertheless, this approach has gained substantial traction in recent years and is now standard practice in a few medical institutions [7].

Despite the proven benefits, the insufficient implementation of ERAS represents missed opportunities for patients. Previous studies have identified barriers to ERAS implementation within healthcare facilities. Notably, organizational constraints and resource availability [20,21,22,23,24,25,26], resistance to change [20,21,22,25,26] and individual variability [21,23,24,27] have emerged as primary barriers. ERAS involves the collaboration of both medical and paramedical staff, although earlier investigations have predominantly focused on medical professionals, particularly surgeons and anesthesiologist [21,22,24,25,26,27,28], with less emphasis on paramedical staff, primarily nurses [21,22,25,26,27,28]. Only two studies have involved paramedical professionals such as physiotherapists or dietitians [26,27]. Notably, none of these prior studies have explored outpatient healthcare providers, despite the applicability of ERAS to both inpatient and outpatient care [7]. Consequently, there exists a critical need to investigate barriers to ERAS implementation across a wide spectrum of healthcare providers, including those in hospitals and outpatient settings. Bridging this knowledge gap would facilitate the development of tailored strategies to overcome these barriers.

Hence, this study aims to address the following questions:What proportion of French healthcare providers actively practice ERAS for major surgery?What are the barriers to ERAS implementation among healthcare providers, both those who practice ERAS and those who do not?What disparities exist between participants who practice ERAS and those who do not?

## 2. Method

### 2.1. Study Design

We conducted a prospective cross-sectional study in accordance with STROBE guidelines [29] (Appendix A). Given the survey’s nature regarding professional practices in healthcare providers (exception II-2, Article R1121-1), no ethical approval was necessary [30]. French healthcare providers were surveyed regarding their practice of ERAS through a questionnaire (Appendix A). Participants were recruited through hospital requests (87.88% from each chief town of French departments), requests to the largest private hospitals (the seven largest private groups in France), professional corporation requests (encompassing 100.00% of professional corporations representing healthcare providers), social networks (public posts and through four different professional groups on two different social networks), and personal contacts. Data collection occurred between December 2020 and February 2021. Some hospitals refused to distribute our questionnaire or were unreachable. A new request was performed each month. We pre-specified that the survey would conclude when the desired sample size was reached, or when the response rate did not increase by at least 5% after a request, whichever came first.

The questionnaire consisted of two parts. The first part assessed whether participants practiced ERAS or not, while the second part was designed to explore barriers to ERAS implementation. The latter was structured as multiple-choice questions, considering factors previously identified as potential barriers [1]. Participants were also given the opportunity to elaborate on any additional barrier via open-text responses. Nine independent healthcare providers tested the questionnaire initially, and minor adjustments were made at this stage, primarily focusing on wording and adding open-ended questions. A table of characteristics of the pretest population is available in Appendix A. During this phase, the estimated time to complete the questionnaire was approximately 6 min 00 s (IQR 6 min 00 s to 6 min 45 s).

### 2.2. Participants

Eligible participants included both graduated medical healthcare providers (anesthesiologists, general physicians, medical specialists, and surgeons) and paramedical healthcare providers (dieticians, nurses, nurse anesthetists, nursing assistants, occupational therapists, operating room nurses, physiotherapists, private nurses, and psychologists). All levels of medical providers, including junior doctors, senior doctors, and specialists, were included. Participants had to work in France, either as self-employed and/or as employees in private or public facilities. Participants whose first language was not French were not included.

Participants who did not complete the questionnaire in its entirety were excluded.

### 2.3. Primary Outcome

The primary outcome was the proportion of participants practicing ERAS for major surgery in France.

To determine major surgery [31], we used the Earl criteria [32] as a reference, including procedures requiring general anesthesia, the opening of great body cavities, the risk of hemorrhage, putting the patient’s life at stake, and requiring special anatomical knowledge and manipulative skills.

The proportion of participants who practiced ERAS for major surgery was analyzed through Questions 8, 10, and 11.

In case of missing or inconsistent data on ERAS practice, participants were conservatively considered non-practitioners of ERAS.

### 2.4. Secondary Outcomes

Barriers to ERAS implementation reported by participants were analyzed through a thorough analysis using the Theoretical Domains Framework (TDF) [33,34]. The TDF is a valuable tool for analyzing and compiling data, especially for investigating barriers to the implementation of a particular practice. The TDF encompasses different domains such as “Behavioral regulation”, “Belief about capabilities”, “Belief about consequences”, “Emotion”, “Environmental context and resources”, “Goals”, “Intentions”, “Knowledge”, “Memory, attention, and decision processes”, “Optimism”, “Reinforcement”, “Skills”, “Social and professional role/identity”, and “Social influences”. The different barriers of ERAS implementation were allocated to one of the fourteen TDF domain [33,34] by two independent assessors, based on answers to pre-established questions (Questions 12 and 14) and open-text responses (Questions 13, 15, and 16). Open-text responses were first read and bundled into topics and then allocated to a domain of the TDF. The two assessors based their allocation on a predefined table of definitions for each domain (see Appendix A). Any disagreements were resolved through discussion or, if necessary, by the adjudication of a third author.

For the secondary outcome analysis, participants were required to either practice ERAS in major surgery or not practice ERAS at all. Participants practicing ERAS for minor surgery were excluded from the secondary outcome analysis.

Secondary outcomes involved the proportion of domains identified as potential barrier to ERAS implementation, using the TDF, among all healthcare providers and specifically among those who practiced ERAS and those who did not in major surgery. Only one barrier were added to a domain, if one participant identified several barriers in the same domain of the TDF regarding questions Q12, Q13, and Q16 for those who practiced ERAS and questions Q14, Q15, and Q16 for those who did not practice ERAS. Differences in demographic characteristics (age, gender, profession, country of studies, time from graduation, and mode of practice) and the proportion of TDF domains identified as barriers to ERAS implementation between participants who practiced ERAS and those who did not were also investigated.

### 2.5. Sample Size Calculation

The target population was French healthcare providers according to the predefined list, with a convenience sample chosen. In the absence of data on the proportion of participants practicing ERAS, we conservatively hypothesized that 50% of healthcare providers practiced ERAS from a statistical point of view. A previous study by Redwood et al. estimated that 57% of breast reconstruction followed ERAS protocols [35]. Assuming a 10% total width for the 95% confidence interval, we aimed to recruit a total of 402 participants.

### 2.6. Statistical Analysis

Demographic characteristics are reported as counts (proportions), mean (standard deviation (SD)), or medians (inter-quartile range (IQR)) based on their distribution, as determined by the Shapiro–Wilk normality test.

The primary outcome (the proportion of healthcare providers practicing ERAS) was presented as a proportion with a 95% confidence interval (95%CI).

The agreement between the two assessors for allocating barriers according to the TDF was assessed using a Kappa score. After allocation, each TDF domain was expressed as a proportion (95%CI) of participants identifying this domain as a barrier among all participants.

The baseline characteristics of participants practicing ERAS and those who did not were compared using the chi-square test, Fisher’s test, Wilcoxon test, or Mann–Whitney test. The comparison of ERAS barriers between participants practicing ERAS and those who did not was conducted using a risk ratio (95%CI), along with the chi-square test and Fisher’s test for each TDF.

## 3. Results

### 3.1. Participants

Data collection was concluded after a total period of two months, as there was only a marginal increase in response rate (<5%). A total of one hundred and fifty-three participants completed the questionnaire, and there were no missing data regarding ERAS practice. For the secondary outcome, nine participants were excluded as they practiced ERAS for minor surgery. Consequently, one hundred and forty-four participants were included for the secondary outcomes (Figure 1).

Demographic characteristics are summarized in Table 1. Most participants were female (75.8%) with a median age of 35 years (IQR 28.5 to 47.5). Most participants were paramedical healthcare providers (86.9%), with physiotherapists, nurses, and dieticians being the most represented professions (31.4%, 24.2%, and 14.4%, respectively). The median time since graduation was 12 years (IQR 5.5 to 22.5). Approximately 78.4% of the participants were employed by an institution. A breakdown of the participants’ origin is shown in Figure 2.

### 3.2. Primary Outcome: Proportion of Participants Who Practiced ERAS

A total of 31 participants practiced ERAS for major surgery (20.3%, 95%CI: 13.9 to 26.63).

Participants practiced ERAS in different surgical specialties, including abdominal surgery (64.5%), thoracic surgery (38.7%), and cardiac surgery (9.7%). Four participants (12.9%) practiced ERAS in at least two of these specialties.

### 3.3. Secondary Outcome

#### 3.3.1. Barriers to Implementation

After the first allocation of answers from Questions Q12 to Q15, there was a minimal agreement [36] between the two assessors, resulting in a Kappa score of 0.37. At the second allocation, they reached full agreement (Kappa score: 1). The two assessors allocated barriers into five different domains among the fourteen TDF. Raw data and the allocation of each answer from Questions Q12 to Q15 are available in Appendix A.

In the overall population, the primary barriers hindering ERAS development were related to “Environmental context and resources” (57.6%, 95%CI: 49.5–65.4) and “Knowledge” (52.8%, 95%CI: 44.7–60.8). Barriers to ERAS implementation for healthcare providers are summarized in Table 2. “Environmental context and resources” issues were attributed to a lack of healthcare providers, insufficient dedicated funding, and poor coordination between hospital and private practice. “Knowledge”-related issues were primarily due to a lack of awareness about ERAS, as 43.8% (95%CI: 35.9–51.7) of responders were unaware of ERAS. “Social and Professional role/identity” constituted a significant part of the remaining barriers (20.8%, 95%CI: 15.0–28.2), with participants expressing that they did not feel concerned about this kind of protocol or reported difficulties working in multidisciplinary teams.

Among participants who practiced ERAS, the TDF domain “Environmental context and resources” was the primary barrier (67.7%, 95%CI: 50.1–81.4). The “Intentions” domain also represented a substantial part of barriers (61.3%, 95%CI: 43.8–76.3), with difficulties mainly attributed to resistance to changing care habits. “Social and Professional role/identity” accounted for 22.6% of the participants (95%CI: 11.4–39.8).

For participants who did not practice ERAS, the main barrier was related to issues of “Knowledge” (61.9%, 95%CI: 52.7–70.4). Indeed, 59.3% (95%CI: 50.2–68.4) of the participants who did not practice ERAS were unaware of it, and issues with information and training were also reported. The “Environmental context and resources” domain (54.9%, 95%CI: 45.7–63.7) was a significant part of the barrier, with reported difficulties in exchanging with nearby health facilities or dependence on their institutions. For 20.4% of the participants (95%CI: 14.0–28.7), barriers were related to “Social and Professional role/identity”, as they did not feel concerned about ERAS.

Participants who practiced ERAS were significantly older than those who did not practice it, with respective median ages of 44.0 years (IQR 34.0 to 54.0) and 33.0 years (IQR 27.0 to 43.5), *p* < 0.001. A higher proportion of surgeons practiced ERAS (22.6% compared to 2.7%, *p* = 0.001). Among those who practiced ERAS, the majority worked as employees (96.8%), whereas there was a lower proportion of employees among those who did not practice it (72.6%). The characteristics of participants who practiced ERAS and those who did not are summarized in Table 3.

#### 3.3.2. Differences

The comparison of barrier types between participants who practiced ERAS and those who did not revealed substantial differences in the proportion of reported barriers between the two groups (*p* < 0.0001), as summarized in Table 4. The proportion of barriers in the TDF domain “Knowledge” was significantly higher in those who did not practice ERAS (RR = 3.2, 95%CI: 1.5–6.7, *p* < 0.001), while the proportion in the TDF domain “Intentions” was significantly higher in those who practiced ERAS (*p* < 0.001).

## 4. Discussion

The main finding of this study is that ERAS is seldom used, with only around a fifth of the respondents practicing it. The study established that “Environmental context and resources”, “Knowledge”, and “Social and professional role/identity” were the main barriers to ERAS implementation and that these barriers may vary between those who practiced ERAS and those who did not. This suggests distinct challenges faced by those who practiced ERAS and those who did not.

Although the estimate of ERAS practice proportion comes with some uncertainty, even based on the upper bound of the 95% CI, it appears that only a quarter of healthcare providers practice ERAS, which remains low. This finding suggests that there is considerable room for improvement in the implementation of ERAS principles in the French healthcare system. Furthermore, it is worth noting that the proportion of healthcare providers who practiced ERAS might even overestimate the true prevalence due to the possibility of omission bias, a common phenomenon in surveys. Omission bias occurs when respondents are less likely to participate in a survey regarding an unfamiliar subject, which could potentially skew the results. On the contrary, it is common for healthcare facilities to adopt some of these principles without a comprehensive ERAS protocol [24]. Consequently, if we focus specifically on the implementation of individual ERAS principles, such as preoperative training, morphine sparing, minimally invasive surgery, early oral intake, or early mobilization, the proportion might be higher. Additionally, our study’s sample composition primarily consisted of paramedical professionals who may not be consistently involved in ERAS protocols, in contrast to hospital medical staff who are more likely to be engaged in ERAS implementation. These varying sources of potential misestimation emphasize the need for further research to comprehensively assess the extent of ERAS implementation, taking into account the following settings: omission bias, the individual implementation of ERAS principles, and the sample composition.

Regarding the barriers of ERAS implementation, our study highlights three main barriers of the TDF, an interesting tool to systematically identify and categorize these barriers. Some of the challenges reported in earlier research, which did not use this tool, align with the domains identified in our study, particularly related to the “Environmental context and resources” domain. For instance, these previously identified barriers include inadequacies in human and material resources [20,21,22,23,24,26,27], communication gaps [20,21,24], and coordination issues between healthcare providers [22,24,25,26,28].

In this study, it is evident that a significant part of the barriers assigned to the “Environmental context and resources” TDF domain could be attributed to different actors such as institutions, healthcare providers, or patients. The previous work of Thompson et al. divided barriers according to these three actors of ERAS implementation. However, most barriers of each actor group refer to the “Environmental context and resources” domain within the TDF [37], which is the most prevalent barrier in the current study. Institutions, the most blamed actor in our study, has been described as a driving force in successful ERAS implementation [37]. They can offer essential organizational support necessary for evidence-based practice like ERAS [38]. Conversely, smaller private practices may lack the capacity to provide such support, potentially impeding ERAS adoption outside of institutional settings. Consequently, the groups previously described can be used as subgroups of this domain to better describe the barriers and suggest more actor-targeted action to overcome the barriers, with a keen focus on institutions.

Moreover, within “Environmental context and resources” issues, our findings highlight a concerning gap in communication and coordination between hospital healthcare providers and primary care professionals. Participants reported difficulties in addressing patients at the time of hospital discharge, because of late requests to primary care providers. Effective coordination between these two actors is necessary for enhancing the quality of care [39], as a multidisciplinary coordination is an important key of the successful implementation of ERAS [40]. Some initiatives in interprofessional collaboration like interprofessional rounds, meetings, or checklist at the local level may help overcome this gap [41]. The development of healthcare networks demonstrates their willingness to improve the uptake of evidence-based practice [42]. Additionally, seminars and national leaders could be supportive [43].

The development of advanced technologies for telehealth might be also useful in the perioperative stage to alleviate some resource-related issues. Indeed, a home rehabilitation program might support the problem of care offer in the preoperative stage but has to be safe and feasible. Several options are currently being studied: for example, a smartphone app involving perioperative recommendations and chest physical exercises [44] or a tele-rehabilitation program with aerobic training [45]. To reduce resource-related barriers, such as those shown in our study, home tele-rehabilitation might help in reducing transportation costs and increasing the accessibility of healthcare services in rural areas or underserved areas.

A gap in “Knowledge”, identified in this study, was also previously reported in the literature with a lack of well-established ERAS protocols [24]. Some of the ERAS principles may be widely used by healthcare providers like minimally invasive surgery [24] whereas others are less commonly used like early oral intake [46]. This can explain the low rate of knowledge about ERAS in our study whereas some principles are well known and applied. Interestingly, healthcare providers with a better understanding of ERAS were more likely to set up ERAS principles [46], suggesting a potential role of education as an intervention to improve ERAS practice. Springer et al. were interested in the training of surgeons, and most of them were trained during their minimally invasive surgery or colorectal fellowship [24]. Initial training in ERAS, through university and training institution support, is then an interesting way to train and inform healthcare providers [43]. The current study also showed that participants who did not practice ERAS, did not know ERAS. However, a lack of experienced and trained professionals was previously described as a barrier [37]. To alleviate this gap, a directly “on the job” training can be used and has shown promising results [43]. Poor ERAS participation due to barriers in “Knowledge” has already been shown [37,43]. In a survey involving 223 participants, Beal et al. also identified knowledge as an important barrier, though this barrier was less frequent, as compared with our findings, with a higher level of knowledge, with only 30% who were unaware of ERAS [43], whereas our study showed that almost 50% of the participants did not know ERAS. This difference could be due to the sample composition. Indeed, in our study, most of the participants were paramedical healthcare providers whereas surgeons and anesthesiologists composed the sample of Beal et al. [43]. Paramedical staff is probably less informed about ERAS than medical staff. Initial training probably has an important role in informing future healthcare providers about ERAS.

Intergroup analysis sheds light on the differences between those who practiced ERAS and those who did not. Participants who practiced ERAS were more experienced, e.g., older and graduate for longer, than those who did not practice ERAS. Participants who practiced ERAS were mainly employees of a healthcare facility. Moreover, participants who did not practice ERAS did not practice it because they had never heard about it. ERAS training is also part of the continuous education of healthcare providers but the learning context differs according to the mode of professional practice. As access to peers, funding, management support, and governance structures are enablers to facilitate learning [47], it is easier to train and implement ERAS with those who worked as salaried healthcare providers. Due to the lack of self-employed healthcare providers in the sample of this study, we cannot draw any conclusions about the difficulties encountered by this group in practicing ERAS. Nevertheless, it could be interesting for further studies to focus on barriers of the implementation of ERAS among self-employed healthcare providers.

### Strengths and Limitations

This study was built on a thorough analysis based on the TDF. The fourteen domains were shown to be effective in analyzing behaviors and improving healthcare outcomes [33]. The domain attribution was led by two independent assessors. This survey is amongst the first to assess the barriers of implementation in a large spectrum of health providers comprising a high rate of paramedical staff and from different modes of practice (employees, self-employed or both).

However, this study also has certain limitations. It focuses on the practice and barriers of ERAS implementation in a single, high-income country, employing a convenient sample. The sample size was not achieved as initially anticipated, potentially affecting the comprehensive identification of barriers and their relative weights. As for representativeness, the convenience of the sample and the response rate constraints limit the generalizability of our findings to all French healthcare providers. Future research might consider using a stratified sample per profession and halt inclusion after reaching a representative threshold. The unanticipated low response rate may be attributed to a lack of interest in the subject [48], influenced by the specifics of working hours and availability constraints of the target population [49]. The amount of data used to link back to the TDF are also limited, as shown by the Kappa score in the first round of assignment. Addressing these limitations will be essential for further research in this area. For instance, further research should include some semi-structured interviews and allocate answers to TDF at three levels: institutions, healthcare providers, and patients.

## 5. Conclusions

In conclusion, the practice of Enhanced Recovery After Surgery (ERAS) remains limited, primarily due to an unfavorable work environment with inadequacies in available resources but also due to a gap in ERAS awareness. To overcome these challenges and promote the wider implementation of ERAS, it might be interesting for future studies and initiatives to focus on implementing strategic solutions. These measures may include the establishment of comprehensive patient management that overcomes the gap between hospital and private care settings, widespread informational campaigns targeting both salaried and self-employed healthcare providers, the integration of ERAS education into the initial training of healthcare professionals, and collaborative efforts involving public authorities, institutions, universities, training institutes, associations, and experienced professionals. These actions are necessary for unlocking the potential of ERAS and improving its implementation across the healthcare landscape.

## Figures and Tables

**Figure 1 healthcare-12-00436-f001:**
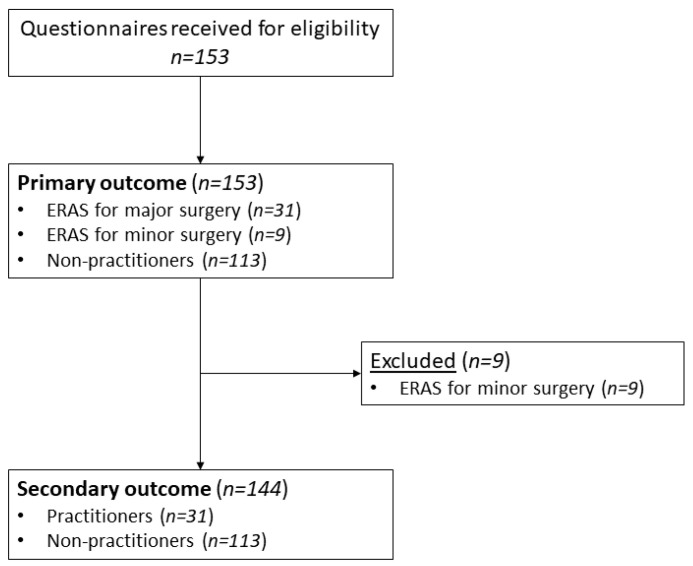
Participant flowchart.

**Figure 2 healthcare-12-00436-f002:**
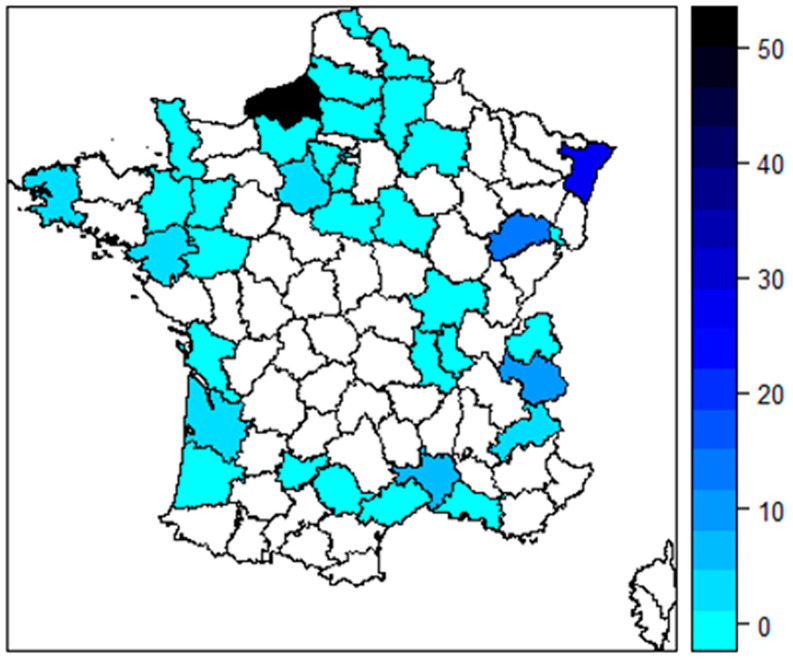
Map of the participants’ origin in France.

**Table 1 healthcare-12-00436-t001:** Characteristics of the participants.

Variable	Global, n = 153
Age (Years), median (IQR)	35 (28.5–47.5)
Gender n (%)	
Female	116 (75.8%)
Profession, n (%)	
Physiotherapist	48 (31.4%)
Nurse	37 (24.2%)
Dietician	22 (14.4%)
Surgeon	12 (7.8%)
Occupational therapist	11 (7.2%)
Psychologist	7 (4.6%)
Anesthesiologist	5 (3.3%)
Nurse anesthetist	3 (2.0%)
Operating room nurse	3 (2.0%)
General physician	2 (1.3%)
Nursing assistant	2 (1.3%)
Medical specialist	1 (0.7%)
Private nurse	0 (0.0%)
Country of studies n (%)	
France	140 (91.5%)
Germany	6 (3.9%)
Belgian	5 (3.3%)
Spain	2 (1.3%)
Time from graduation (Years), Median (IQR)	12 (5.5–22.5)
Practice mode, n (%)	
Employee	120 (78.4%)
Self-employed	23 (15.0%)
Mixed	10 (6.5%)

**Table 2 healthcare-12-00436-t002:** Barriers to ERAS in healthcare providers.

TDF *	Global, n = 144	Global, % (95%CI)
Environmental context and resources	83	57.6% (49.5–65.4)
Knowledge	76	52.8% (44.7–60.8)
Social and professional role/identity	30	20.8% (15.0–28.2)
Intentions	19	13.2% (8.6–19.7)
Belief about consequences	7	4.9% (2.4–9.7)

* Only the domains reported by respondents have been included in the table.

**Table 3 healthcare-12-00436-t003:** Characteristics and comparison of participants who practiced ERAS and those who did not.

Variable	Participants Who Practiced ERAS, n = 31	Participants Who Did Not Practice ERAS, n = 113	*p*-Value
Age (Years), Median (IQR)	44.0 (34–54)	33.0 (27.0–43.5)	<0.001 *
Gender, n (%)			0.096
Female	20 (65.0%)	90 (80.0%)	
Profession, n (%)			<0.001 *
Physiotherapist	9 (29.0%)	35 (31.0%)	>0.999
Nurse	6 (19.4%)	29 (25.7%)	0.637
Dietician	3 (9.7%)	19 (16.8%)	0.410
Occupational therapist		11 (9.7%)	
Surgeon	7 (22.6%)	3 (2.7%)	0.001 *
Psychologist		7 (6.2%)	
Anesthesiologist	3 (9.7%)	1 (0.9%)	0.031 *
Nurse anesthetist		3 (2.7%)	
Operating room nurse	3 (9.7%)		
General physician		2 (1.8%)	
Nursing assistant		2 (1.8%)	
Medical Specialist		1 (0.9%)	
Private nurse			
Country of studies, n (%)			0.846
France	28 (90.3%)	105 (92.9%)	0.703
Germany	2 (6.5%)	4 (3.5%)	0.610
Belgian	1 (3.2%)	3 (2.7%)	>0.999
Spain		1 (0.9%)	
Time from graduation, Median (IQR)	15 (8–29)	11 (4.5–21)	0.052
Practice mode, n (%)			0.016 *
Employee	30 (96.8%)	82 (72.6%)	0.003 *
Self-employed	1 (3.2%)	22 (19.5%)	0.028 *
Mixed		9 (8.0%)	

* *p* < 0.05.

**Table 4 healthcare-12-00436-t004:** Barriers to ERAS implementation in participants who practiced ERAS and those who did not.

TDF	Participants Who Practiced ERAS, n = 31	Participants Who Did Not Practice ERAS, n = 113	Relative Risk (95%CI)	*p*-Value
Environmental context and resources, % (95%CI)	67.7% (50.1–81.4)	54.9% (45.7–63.7)	0.8 (0.6–1.1)	0.224
Knowledge, % (95%CI)	19.4% (9.2–36.3)	61.9% (52.7–70.4)	3.2 (1.5–6.7)	<0.001 *
Social and professional role/identity, % (95%CI)	22.6% (11.4–39.8)	20.4% (14.0–28.7)	0.9 (0.4–1.9)	0.805
Intentions, % (95%CI)	61.3% (43.8–76.3)	0.0% (0.0–0.0)		<0.001 *
Belief about consequences, % (95%CI)	16.1% (7.1–32.6)	1.8% (0.5–6.2)	0.1 (0.2–0.5)	0.005 *

* *p* < 0.05.

## Data Availability

Data are contained within the article and Appendix A.

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
