# Peer review of "Enhanced Recovery after Surgery (ERAS) Implementation and Barriers among Healthcare Providers in France: A Cross-Sectional Study"

_healthcare, 2024, doi:10.3390/healthcare12040436_

Round 1

Reviewer 1 Report (Previous Reviewer 4)

Comments and Suggestions for Authors

Delete the “.” At the end of title

Line 31, “Median age”, you meant mean age? If so what about the SD?

Comments on the Quality of English Language

extensive

Author Response

Delete the “.” At the end of title

According to your comments the point was deleted.

Line 31, “Median age”, you meant mean age? If so what about the SD?

We used median and IQR (interquartile ratio) to define age. Indeed, distribution was tested by Kolmogorov-Smirnov test (KS distance = 0.1142 ; p value <0.0001) and the distribution of the variable “age” is non-parametric. Consequently, we assumed that median and IQR are more relevant than mean and SD in this case.

Reviewer 2 Report (Previous Reviewer 2)

Comments and Suggestions for Authors

Overall, a fairly well written paper. The introduction and methods are significantly improved from the previous submission. Well done. There are limitations mostly within your results section, as outlined below. 

The results would benefit from more explanation around how you aligned the findings with the TDF. For example, which answers to Q12 and Q14 were aligned with which TDF domain? How did you determine this? Are there any supporting quotes in the open comments that can support your findings? I would like to see the results of the survey questions 12 and 14, not just your interpretation. 

Discussion: Do you think the answers to 2 questions about barriers are robust enough to give good translatable alignment?

The limitations you have discussed are valid, but I would also include the limited amount of data that you used to link back to the TDF. 

Figure 1 belongs in the methods section. 

Comments on the Quality of English Language

The language is fine - some minor errors. 

Author Response

Overall, a fairly well written paper. The introduction and methods are significantly improved from the previous submission. Well done. There are limitations mostly within your results section, as outlined below.

Thank you for your previous comments that allowed us to improve the quality of manuscript. Please find answers to your questions and comments below.

The results would benefit from more explanation around how you aligned the findings with the TDF. For example, which answers to Q12 and Q14 were aligned with which TDF domain?

Thank you for your comment. Please find below the tables of answers assignment for the questions Q12 to Q15.

Table A: Answers and TDF allocation to question Q12

TDF domain

Q12 – Answers

n (%)

“Environmental context and resources”

o   The regular change of professionals within the department.

14 (45.2%)

o   Lack of funding.

6 (19.4%)

o   Poor city-hospital coordination.

5 (16.1%)

o   The absence of a financing nomenclature adapted to these patients.

4 (12.9%)

o   The increase in the number of administrative tasks.

4 (12.9%)

o   ERAS labels for healthcare facilities.

1 (3.2%)

o   The turnover in the installation of self-employed on the territory.

0 (0.0%)

“Intentions”

o   Changing care habits to adopt new practices.

19 (61.3%)

“Social and professional role / identity”

o   Multidisciplinary teamwork.

7 (22.6%)

Figure A: Answer and TDF allocation to question Q12

Table B: Answers and TDF allocation to question Q13

TDF domain

Q13 – Themes of identified obstacles

n (%)

“Environmental context and resources”

o   Coordination issues within healthcare facilities

2 (6.5%)

o   COVID19 period

1 (3.2%)

o   Staff shortage

1 (3.2%)

o   Funding issues

1 (3.2%)

o   Medico-legal pressure in the event of complications

1 (3.2%)

“Knowledge”

o   Lack of knowledge of ERAS in healthcare facilities

5 (16.1%)

“Belief about consequences”

o   Patient condition

5 (16.1%)

“Skills”

o   Lack of training

2 (6.5%)

“Intentions”

o   Lack of motivation from healthcare providers

1 (3.2%)

Figure B: Answers and TDF allocation to question Q13

Table C: Answers and TDF allocation to question Q14

TDF domain

Q14 – Answers

n (%)

“Knowledge”

o   I had never heard about it.

55 (48.7%)

“Environmental context and resources”

o   I have never seen a patient in a ERAS protocol.

40 (35.4%)

o   The health care facility where I work does not have an ERAS protocol.

25 (22.1%)

o   I have not been contacted by the health care facilities around my practice.

9 (8.0%)

o   The health care facilities around my practice do not ERAS.

8 (7.1%)

o   I don’t have the means to ERAS patient.

3 (2.7%)

o   I don’t have the time to ERAS patient.

0 (0.0%)

“Social and professional role / identity”

o   I do not feel concerned by this type of care.

11 (9.7%)

“Intentions”

o   I do not wish to change my care habits.

0 (0.0%)

Figure C: Answers and TDF allocation to question Q14

Table D: Answers and TDF allocation to question Q15

TDF domain

Q15 – Themes of identified obstacles

n (%)

“Social and professional role / identity”

o   Not suitable for the care unit or workplace

13 (11.5%)

o   Not suitable for my profession

1 (0.9%)

“Environmental context and resources”

o   Non-ERAS-labelled protocol

4 (3.5%)

o   Poor city-hospital coordination

2 (1.8%)

o   Refusal of hierarchy

1 (0.9%)

“Knowledge”

o   Lack of information

3 (2.7%)

“Belief about consequences”

o   ERAS failure

1 (0.9%)

o   Need of invasive surgery

1 (0.9%)

Figure D: Answers and TDF allocation to question Q15

A figure with requested details was added to the Supplemental data as Appendix 5.

3.3. Secondary Outcome

3.3.1. Barriers to Implementation

After the first allocation of answer of question Q12 to Q15, there was a minimal agreement (36) between the two assessors, resulting in a Kappa score of 0.37. At the second allocation, they reached full agreement (Kappa score: 1). The two assessors allocated barriers into five different domains among the fourteen TDF. Raw data and allocation of each answer from Question Q12 to Q15 are available in Appendix 5.”

How did you determine this?

Q12 and Q14:

A list of definition of each domain of the TDF was shared to the 2 independent assessors. Each possibility of answer was assigned independently by 2 assessors to one domain of the TDF for question Q12 and Q14.

Q13 and Q15:

For questions Q13 and Q15, answers were bundled into themes. These themes were assigned by 2 independent assessors to one domain of the TDF. For instance, one participant specified at question Q13 “certains patients venus en urgence refusent......” [Translation: some emergency patients refuse......], the theme extracted was then “patient condition” and the assigned TDF was “Belief about consequences”.

Further details of TDF allocation were added in the method part. Please see below :

“2.4 Secondary Outcomes

Barriers to ERAS implementation reported by participants were analyzed through a thorough analysis using the Theorical Domains of Framework (TDF) (33,34). TDF is valuable tool for analyzing and compiling data, especially for investigating barriers to the implementation of a particular practice. The TDF encompasses different domains such as “Behavioral regulation”, “Belief about capabilities”, “Belief about consequences”, “Emotion”, “Environmental context and resources”, “Goals”, “Intentions”, “Knowledge”, “Memory, attention, and decision processes”, “Optimism”, “Reinforcement”, “Skills”, “Social and professional role/identity”, and “Social influences”. The different barriers of ERAS implementation were allocated to one of the fourteen TDF domain (33,34) by two independent assessors, based on answers to pre-established questions (Questions 12 and 14) and open-text responses (Questions 13, 15 and 16). Open-text responses were first read and bundled into topics and then allocated to a domain of the TDF. The two assessors based their allocation on a predefined table of definitions for each domain [see Appendix 4]. Any disagreements were resolved through discussion or, if necessary, by adjudication of a third author.

Are there any supporting quotes in the open comments that can support your findings?

Answers to the open comments were added to the analysis as seen before.

I would like to see the results of the survey questions 12 and 14, not just your interpretation.

Thanks to your comment, some explanations were added in method and results part. Answers to Q12 to Q14 are expressed in the table above. In these tables results are expressed in number of participants who choose this answer. For our analysis, we choose to express the results in number of participants who identified at least one answer in the relevant domain. Indeed, if one participant chose several answer assigns to the same domain of TDF, this will be count for only one.  For instance, a patient identified “Lack of funding” and “Poor city hospital coordination” as obstacle to ERAS implementation, then we only added 1 in the domain “Environmental context and resources”.

According to your comments the manuscript was amended, please see below :

“2.4 Secondary Outcomes

[…]

Secondary outcomes involved the proportion of domains identified as potential barrier to ERAS implementation, using the TDF, among all healthcare providers and specifically among those who practiced ERAS and those who did not in major surgery. Only one were added to a domain, if one participant identified several barriers in the same domain of the TDF regarding questions Q12, Q13 and Q16 for those who practiced ERAS and questions Q14, Q15 and Q16 for those who didn’t practice ERAS. Differences in demographic characteristics (age, gender, profession, graduate country, time from graduation and mode of practice) and the proportion of TDF domains identified as barriers to ERAS implementation between participants who practiced ERAS and those who did not were also investigated.”

Discussion: Do you think the answers to 2 questions about barriers are robust enough to give good translatable alignment?

Indeed, you’re right. Some semi-structured interviews will be more appropriate to assess the different obstacles of ERAS. Moreover, for future studies, I think it will be more appropriate to explore three main actors through TDF, i.e. institutions, healthcare providers and patients to develop specific strategies in each domain to encompass the identified obstacles.

The limitations you have discussed are valid, but I would also include the limited amount of data that you used to link back to the TDF.

Thank you for your suggestion. We agree the amount of data are limited to assign a domain of the TDF as shown by the Kappa score at the first round of assignment (Kappa score = 0.37). Indeed, assessors based their assignment on a definition. According to your comment, the manuscript has been revised.

4.1 Strengths and Limitations

[…]

However, this study has also certain limitations. It focuses on the practice and barriers of ERAS implementation in a single, high-income country, employing a convenient sample. The sample size was not achieved as initially anticipated, potentially affecting the comprehensive identification of barriers and their relative weights. As for representativeness, the convenience sample and the response rate constraints limit the generalizability of our findings to all French healthcare providers. Future research might consider use a stratified sample per profession and halt inclusion after reaching a representative threshold. The unanticipated low response rate may be attributed to a lack of interest in the subject (48), influenced by the specific of work hours and availability constraints of the target population (49). The limited amount of data used to link back to the TDF is also limited as shown by the Kappa score at the first round of assignment. Addressing these limitations will be essential for further research in this area. For instance, further research should include some semi-structured interviews and allocate answer to TDF at three levels: institutions, healthcare providers and patients.

Figure 1 belongs in the methods section.

Thank you for your attention, the figure was nevertheless moved to the result section as it comprises some answers to the first outcome.

Reviewer 3 Report (Previous Reviewer 1)

Comments and Suggestions for Authors

No other comments

Author Response

More explanation about the research design were added according to your suggestion. Please find below the modifications : 

"2.4. Secondary Outcomes

Barriers to ERAS implementation reported by participants were analyzed through a thorough analysis using the Theorical Domains of Framework (TDF) (33,34). TDF is valuable tool for analyzing and compiling data, especially for investigating barriers to the implementation of a particular practice. The TDF encompasses different domains such as “Behavioral regulation”, “Belief about capabilities”, “Belief about consequences”, “Emotion”, “Environmental context and resources”, “Goals”, “Intentions”, “Knowledge”, “Memory, attention, and decision processes”, “Optimism”, “Reinforcement”, “Skills”, “Social and professional role/identity”, and “Social influences”. The different barriers of ERAS implementation were allocated to one of the fourteen TDF domain (33,34) by two independent assessors, based on answers to pre-established questions (Questions 12 and 14) and open-text responses (Questions 13, 15 and 16). Open-text responses were first read and bundled into topics and then allocated to a domain of the TDF. The two assessors based their allocation on a predefined table of definitions for each domain [see Appendix 4]. Any disagreements were resolved through discussion or, if necessary, by adjudication of a third author.

For the secondary outcomes analysis, participants were required to either practice ERAS in major surgery or not practice ERAS at all. Participants practicing ERAS for minor surgery were excluded from the secondary outcomes analysis.

Secondary outcomes involved the proportion of domains identified as potential barrier to ERAS implementation, using the TDF, among all healthcare providers and specifically among those who practiced ERAS and those who did not in major surgery. Only one were added to a domain, if one participant identified several barriers in the same domain of the TDF regarding questions Q12, Q13 and Q16 for those who practiced ERAS and questions Q14, Q15 and Q16 for those who didn’t practice ERAS. Differences in demographic characteristics (age, gender, profession, graduate country, time from graduation and mode of practice) and the proportion of TDF domains identified as barriers to ERAS implementation between participants who practiced ERAS and those who did not were also investigated."

This manuscript is a resubmission of an earlier submission. The following is a list of the peer review reports and author responses from that submission.

Round 1

Reviewer 1 Report

Comments and Suggestions for Authors

Congratulations for your article. Even if it's talking about and old topic, ERAS in fact is insufficiently implemented.

Comments:

-  to short an introduction: elaborate

- what is the proportion of each profession you have contacted; there is a very small number of doctors who have answer your questionnaire 

Reviewer 2 Report

Comments and Suggestions for Authors

Overall, a potentially interesting study. There are some large gaps in the writing that need to be revised. Please see specific comments below. 

I would remove the short title from the front page

Headings can be removed from the abstract

Line 13 - Why is the implementation a challenge for patients? I would think that the challenge lies with health care providers? 

Line 16 - Barries to use? Or implementation? Or development? 

Line 17 - What do you mean by "involved in"? 

Line 20 - remove "to"

Line 20 - Survey or questionnaire? Be consistent. 

Line 26 - dont start a sentence with a number. It needs to be written in full

Line 33 -Language "is little practiced" needs revising. 

Line 47 - You talk about surgery, but there is no definition. Do you mean all types of surgery? All disciplines? What type of changes has surgery undergone? This is a very broad statement.

Line 49 - you have suggested two challenges but then discuss 4 issues here. 

Table 2 - Why have you not included all domains from the TDF? A better methodological description would help here. 

Line 185 - (and elsewhere) What is an ERAS practitioner? This is not defined anywhere. 

Results - Where are the results of the questions that you asked? How did you determine the alignment with the TDF domains? 

Discussion - You need to exapand on your own results, discuss them at a high level and discuss their relevance. Some of your discussion should be included in the results. You have intriduced recommendations and solutions without evidence (telehealth?) or was this part of your results? 

Line 275 - You have suggested a thorough analysis using TDF but I do not see this in your paper. 

Conclusion - You continue to use the same words "environmental context and resources" and "knowledge" without much meaning behind them. I would suggest re-writing how you are framing your findings. 

References - the reference list is inconsistent. Please revise. 

Line 70 - Your first research question is not robust. How can you extrapolate the proportion of all French healthcare providers based on your sample? 

Line 78 - Why was ethical approval not needed? 

Line 90 - Please describe the TDF appropriately. Why was this tool used? How was it used? You need to include a more detailed description here. TDF is primarily used for qualitative research. 

Line 99 - Did you include all levels of medical staff (junior doctors, senior doctors, specialists)? 

Comments on the Quality of English Language

The language is quite poor and should be reviewed and revised. 

Reviewer 3 Report

Comments and Suggestions for Authors

Greetings,

Thank you very much for this opportunity to read your work. This paper aims to assess the proportion of French healthcare providers involved in ERAS protocols and identify barriers to their development and practice among those surveyed. Overall, The ERAS protocol has been assessed in several studies around the world, and then authors need to justify their topic in a more appropriate way.

1. The study needs to justify its contributions as "theoretical and practical". The study gaps are totally confused and should be extended.

2. Authors need to carefully review the current literature on the topic and then find its gaps. Kindly write a separate section related to the current literature and how your study will contribute to it.

3. Methodology: Further justifications are required on: research strategy, approach, data collection methods, sampling technique, and exclusion criteria. Furthermore, how do authors deal with non-response bias? How do the authors assure the study's reliability and validity? Justification on the data analysis tools used; for example, why did they not use binary logistic regression?

4. Findings: authors need to rewrite the section and link their findings to current literature. Also, theoretical and practical contributions are needed.

Comments on the Quality of English Language

Moderate editing of English language required

Reviewer 4 Report

Comments and Suggestions for Authors

Thank you for submitting your manuscript to Healthcare. You can see my comments below.

·        Short title can be deleted

Abstract

·        “Purpose’? is it “introduction/background”?

·        ERAS should be in full term at the first time

·        Wrong word using “real word” I think you meant “real world”

·        ERAS is a challenge for patients undergoing surgery. What types? It’s better to identify the most concerned surgeries or the areas the authors want to focus. Otherwise, the Introduction may not be focused, and the knowledge gap may not be valid. It affects the application of your finding in the current practice.

·        Healthcare providers were included. Where were they from? What expertise they were at?

·        Results: 153 should be in full English instead of numbers at the beginning of a sentence.

·        What analytic methods were conducted to find the barriers?

The list of abbreviations is not needed in this journal.

Main context

·        This session is too short. The introduction should include the ERAS introduction, what are its importance. What are the problem and why this study had to be done? However, all of this information is inadequate.

·        The aim is not clear

·        What about the healthcare who had no experience of using ERAS? Were they included? If so, did they receive any training to use ERAS before completing the survey?

·        Any validation was done on the tool? What about the reliability? Any pilot study was conducted to test the feasibility of the tool?

·        It’s not clear how to select the healthcare providers. Any inclusion and exclusion criteria? How were they recruited?

·        Primary outcome and secondary outcome were incorrect. The authors should stick with the study aim as mentioned in the Title.

·        Due to different healthcare providers are involved, to achieve the study aim, the sample size calculation should not be considered as one sample. Individual healthcare disciplines should reach a certain number for the credibility of the study.

·        What analytic tests were used to identify the barriers?

Results:

·        What is IQR?

·        Again, the primary and secondary outcomes are incorrect.

·        What about the p values for the barriers?

Discussion

·        Not specific enough. The ideas are fragmented and superficial. This part lacks organization. The authors should interpret the results and discuss how the finding impact the current practice.

·        It is confusing at the description of telehealth and the ERAS as well as its barriers. If telehealth is used, the barriers will be disappeared?

Comments on the Quality of English Language

English proficiency editing is needed.